# Higher SARS-CoV-2 Spike Binding Antibody Levels and Neutralization Capacity 6 Months after Heterologous Vaccination with AZD1222 and BNT162b2

**DOI:** 10.3390/vaccines10020322

**Published:** 2022-02-17

**Authors:** Brigitte Müller-Hilke, Franz Mai, Michael Müller, Johann Volzke, Emil C. Reisinger

**Affiliations:** 1Core Facility for Cell Sorting and Cell Analysis, Rostock University Medical Center, 18055 Rostock, Germany; franz.mai@uni-rostock.de (F.M.); michael.mue@t-online.de (M.M.).; johann.volzke@med.uni-rostock.de (J.V.); 2Division of Tropical Medicine and Infectious Diseases, Center of Internal Medicine II, Rostock University Medical Center, 18055 Rostock, Germany; emil.reisinger@uni-rostock.de

**Keywords:** SARS-CoV-2, COVID-19, heterologous prime-boost vaccination, variant of concern, humoral and cellular immune response

## Abstract

Within a year after the emergence of SARS-CoV-2, several vaccines had been developed, clinically evaluated, proven to be efficacious in preventing symptomatic disease, and licensed for global use. The remaining questions about the vaccines concern the duration of protection offered by vaccination and its efficacy against variants of concern. Therefore, we set out to analyze the humoral and cellular immune responses 6 months into homologous and heterologous prime-boost vaccinations. We recruited 190 health care workers and measured their anti-spike IgG levels, their neutralizing capacities against the Wuhan-Hu-1 strain and the Delta variant using a surrogate viral neutralization test, and their IFNγ-responses towards SARS-CoV-2-derived spike peptides. We here show that IFNγ secretion in response to peptide stimulation was significantly enhanced in all three vaccination groups and comparable in magnitude. In contrast, the heterologous prime-boost regimen using AZD1222 and BNT162b2 yielded the highest anti-spike IgG levels, which were 3–4.5 times more than the levels resulting from homologous AZD1222 and BNT162b2 vaccination, respectively. Likewise, the neutralizing capacity against both the wild type as well as the Delta receptor binding domains was significantly higher following the heterologous prime-boost regimen. In conclusion, our results suggest that mixing different SARS-CoV-2 vaccines might lead to more efficacious and longer-lasting humoral protection against breakthrough infections.

## 1. Introduction

In December 2019, severe acute respiratory syndrome coronavirus 2 (SARS-CoV-2) emerged, causing the coronavirus disease 2019 (COVID-19) pandemic. COVID-19 is a disease of the lower respiratory tract that is characterized by dysregulated inflammatory responses leading to pneumonia, vascular insults, and multi-organ failure [1]. Specific risk factors for severe disease and fatal outcomes are older age; male sex; and several comorbidities such as diabetes, hypertension, coronary heart disease, chronic obstructive pulmonary disease, and other lung pathologies [2,3]. Within two years, the pandemic caused by SARS-CoV-2 led to more than 5 million deaths worldwide [4].

At an unprecedented speed, global endeavors in industry and academia set out to develop vaccines, employing a multitude of well-proven as well as novel technologies. In less than a year, several formulations were approved and licensed, among them the adenoviral-based AZD1222 (Vaxzevria/AstraZeneca, Cambridge, England) and the mRNA-based BNT162b2 (Comirnaty/BioNTech, Mainz, Germany). Both vaccines were shown to be safe and were 95% (BNT162b2) and up to 90% (AZD1222) efficacious against symptomatic COVID-19 when analyzed more than 14 days after the second dose [5,6]. Depending on local licensing and availability, some countries, such as Israel, began by immunizing predominantly with BNT162b2, while others, such as the UK or Germany, initially used homologous prime-boost regimens with either AZD1222 or BNT162b2 [7,8,9]. However, clinical developments very soon led to the application of heterologous prime-boost regimens [10].

As of yet, data are still sparse, or lacking altogether, regarding the most efficient vaccination regimen, the duration of the protection from vaccination, and the effectiveness of both vaccines against the wildtype Wuhan-Hu-1 strain as well as the Delta variant of concern, which accumulated mutations in the spike protein and exhibits a higher transmissibility and replication capacity and causes higher rates of breakthrough infections [11,12]. Therefore, we set out to compare humoral and cellular immune responses 6 months into homologous AZD1222 prime-boost, homologous BNT162b2 prime-boost, and heterologous AZD122/BNT162b2 prime-boost regimens among health care workers in a German medical university center.

## 2. Materials and Methods

### 2.1. Study Participants and Blood Sampling

Study participants included health care personnel, administrators, scientists, and craftspeople at Rostock University Medical Center and were recruited from the local coordination center for clinical studies. Among 190 participants, there were two of Asian descent and one of African descent, while the majority of the participants were of European descent. Apart from their age and sex, no personal information such as health status (i.e., underlying conditions) were documented. Blood samples were obtained by venipuncture 6 months after the first vaccination against SARS-CoV-2. Peripheral blood mononuclear cells (PBMCs) were isolated from anti-coagulated blood by density gradient centrifugation using Ficoll-Paque^TM^ PLUS according to the manufacturer’s instructions (Cytiva, Marlborough, MA, United States). PBMCs were subsequently suspended in fetal calf serum (FCS, Thermo Fisher, Waltham, MA, United States) containing 10% dimethyl sulfoxide (Sigma-Aldrich, St. Louis, MO, United States) and frozen at −70 °C until further use. Plasma samples were obtained by the centrifugation of anti-coagulated blood and were frozen at −70 °C afterwards. Likewise, serum samples were obtained from coagulated blood and frozen again afterwards. This study was approved by the ethics committee of the Rostock University Medical Center under the file number A 2020-0086. Written informed consent was provided by all participants.

### 2.2. SARS-CoV-2 RBD- and Nucleocapsid Specific Antibodies

Serum samples were thawed and tested for the presence of antibodies against the SARS-CoV-2 nucleocapsid or RBD domain of the spike protein via electro-chemiluminescence immunoassay (ECLIA). To that end, the quantitative anti-SARS-CoV-2S and anti-SARS-CoV-2N Elecsys^®^ Assays (Roche Diagnostics, Mannheim, Germany) were performed according to the manufacturer’s instructions and were run on a Cobas E411 (Roche Diagnostics, Mannheim, Germany). Results were compared to the WHO international standard for anti-SARS-CoV-2 immunoglobulin to obtain binding antibody units (BAU) [13].

### 2.3. Neutralizing Capacity against SARS-CoV-2 RBD Wuhan-Hu-1 and Delta B.1.617.2 Variant

Plasma samples were thawed on ice and centrifuged at 10,000× *g* in order to remove precipitates before 10-fold dilutions were made. For the detection of SARS-CoV-2-neutralizing antibodies, a surrogate virus neutralization test was conducted. This test was previously shown to robustly correlate with conventional virus neutralization tests [14]. Here, we followed the manufacturer’s specifications using RBD peptides for both the Wuhan-Hu-1 strain and the B.1.617.2 (L452R, T478K) variant (Genscript, Piscataway Township, NJ, USA). The absorbance was detected at 450 nm (A450) on the Infinite^®^ 200 automated plate reader (Tecan, Männedorf, Switzerland). The optical densities (OD) of the negative control values were used to calculate inhibition percentage. Results for each individual sample were calculated using the following formula: inhibition = (1 − OD_SAMPLE_/OD_NEG CTRL_) × 100. The WHO Reference Panel for anti-SARS-CoV-2 immunoglobulin (20/268) was used for the calibration and calculation of international units (IU) [13].

### 2.4. Human Interferone Gamma ELISPOT

PBMCs were thawed, centrifuged, and suspended in complete RPMI medium containing 10% FCS, 100U penicillin/ 0.1 mg streptomycin, 2nM L-glutamine (Thermo Fisher), 10 mM HEPES, and 1 mM sodium pyruvate (PAN-Biotech, Aidenbach, Germany). Cell counts were determined cytometrically on the Cytek^®^ Aurora (Cytek Biosciences, Fremont, CA, United States) using 4′,6-diamidino-2-phenylindole (DAPI, Biolegend, San Diego, CA, United States) as a live/dead discriminator. The flow cytometer operated on the SpectroFlo software version 2.2.0.3 (Cytek Biosciences). An RPMI/cell suspension volume equal to five hundred thousand PBMCs was pipetted into a 96-well V-bottom plate and centrifuged for 5 min at 4 °C and 400× *g*. Subsequently, the supernatants were removed by carefully blotting the plate on a paper tissue. Cells were then suspended in 36 µL of complete RPMI medium containing 0.12 nmol peptides from a pool mainly consisting of 15mer sequences with an overlap of 11 amino acids, covering the immunodominant sequence domains of the spike glycoprotein of the SARS-CoV-2 (Genbank MN908947.3, protein QHD43416.1, strain Wuhan-Hu-1) (PepTivator^®^ SARS-CoV-2 Prot_S, Miltenyi Biotec, Bergisch-Gladbach, Germany). Afterwards, PBMCs were transferred into a 96-well enzyme-linked immune absorbent spot (ELISpot) assay plate coated with capture antibodies specific for human interferon gamma (IFNγ) (R&D Systems, Minneapolis, MN, United States). After incubating the cells for 30 min at 37 °C, 164 µL of complete RPMI medium was added to all wells. This was followed by a 23.5 h incubation at 37 °C in a CO_2_ incubator (Binder, Tuttlingen, Germany). The ELISpot assay was then performed according to the manufacturer’s guidelines. The number of IFNγ-producing cells were determined by automated counting using the ImmunoSpot^®^ analyzer operating on the ImmunoSpot^®^ Software version 5.0.9.15 (CTL Europe, Bonn, Germany) following the guidelines for the automated evaluation of ELISpot assays [15].

### 2.5. Statistics

Data were tested for Gaussian distribution using the Kolmogorov–Smirnov test. Data not following a Gaussian distribution—BAUs and % neutralization capacity—were compared using the Kruskal–Wallis test with Dunn’s multiple comparisons test and the Wilcoxon matched pairs signed rank test, respectively (Figure 1 and Figure 2). Correlations were performed using Spearman rank correlation analysis (Table 1, Appendix A). Absolute numbers of IFNγ-producing ELISpots resulting from stimulation with the peptide pool were compared via one-sample t-tests to the mean + 2 × SEM of the corresponding number of IFNγ-producing ELISpots observed in the absence of stimulation. The numbers of responders and non-responders in each cohort were compared via Chi-square tests for independence. Mann–Whitney U tests were performed to compare the binding antibody units (BAU), neutralization capacities, and IFNγ-producing ELISpots between male and female participants. Statistical assays were performed with GraphPad InStat^®^ version 3.10 for Windows (GraphPad Software, San Diego, CA, USA).

## 3. Results

A total of 190 participants were recruited from the local coordination center for clinical studies. Inclusion criteria were (i) complete vaccination against SARS-CoV-2 at six months after primary immunization, (ii) age between 18 and 65 years, and (iii) no previous infection with SARS-CoV-2. A total of 61 of our participants had received homologous prime-boost immunizations with AZD1222 (Vaxzevria/Astrazeneca) at a median interval of 83 days (min 68/max 145), 66 participants had received homologous prime-boost immunizations with BNT162b2 (Comirnaty/Biontech) at a median interval of 84 days (min 74/max 94), and 65 participants had been subject to heterologous prime-boost immunization with AZD1222 followed by BNT162b2 at a median interval of 29 days (min 23/max 36). Blood samples were obtained by venipuncture and serum and the plasma and PBMCs were stored at −70 °C until further use. An initial screening for antibodies against the viral nucleocapsid led to the exclusion of two participants from the heterologous prime-boost group. Table 1 lists the demographic data of the remaining participants, showing an even distribution of age ranges among the three cohorts. However, there was an unbalanced number of male and female participants with the heterologous prime-boot group consisting of only 17.5% men and the BNT162b2 and AZD1222 prime-boost groups consisting of 25.8% and 37.7%, respectively.

### 3.1. Heterologous Prime-Boost Combining AZD1222 and BNT162b2 Resulted in Highest Antibody Levels at Six Months

In our first approach, we quantified the spike-specific IgG antibodies among our study participants as binding antibody units (BAU) calibrated to the WHO standard [13]. There were significant differences among the three groups, with the heterologous prime-boost group featuring the highest median of 2162 BAU/mL (IQR: 1464–3639), followed by the homologous prime-boost groups featuring medians of 695 (BNT162b2) (IQR: 455-1153) and 489.2 BAU/mL (AZD1222) (IQR: 261-725) (Figure 1). By comparing the homologous prime-boost regimens, we found that BNT162b2 vaccination exhibited significantly more spike binding antibodies than AZD1222. Within each of the vaccination groups, male and female participants had comparable levels of anti-spike antibodies. A negative correlation between age and anti-spike antibody levels was observed for the homologous prime-boost group immunized with BNT162b2 (Appendix A).

### 3.2. Heterologous Prime-Boost Combining AZD1222 and BNT162b2 Yielded Highest Neutralization Capacities at Six Months

The individual neutralization capacities were determined via surrogate virus neutralization tests (sVNT) using an RBD peptide representing the Wuhan-Hu-1 strain [14]. The heterologously primed and boosted group yielded the highest neutralization capacity with a median of 94%, followed by the homologous prime-boost groups with medians of 87% (BNT162b2) and 66% (AZD1222) (Figure 2). When correlating the neutralizing capacities to their respective binding antibody units, we found highly significant results in all vaccination groups. However, rank correlation coefficients indicated a strong correlation in the homologous prime-boost BNT162b2 group (r = 0.80), a moderate correlation in the heterologous prime-boost group (r = 0.67), and a weak correlation in the homologous prime-boost AZD1222 group (r = 0.47) (Appendix A). Again, there were no differences between the sexes, but a significant yet weak negative correlation between age and neutralizing capacity in the homologous prime-boost BNT162b2 group (Appendix A).

Next, we analyzed the individual neutralization capacities against an RBD peptide representing the B.1.617.2 (Delta) variant with the L452R and T478K mutations [12]. Again, the heterologous prime-boost group yielded the highest neutralization capacity with a median of 93.8%, followed by the homologous prime-boost BNT162b2 group with a median of 85.1% and the homologous prime-boost AZD1222 group with a median of 64.8% (Figure 2). While pairwise comparisons of the neutralization capacity against Wuhan-Hu-1 and Delta revealed comparable results, with the exception of the heterologous prime-boost group, these differences were rendered insignificant after correction for multiple comparisons. Correlations between anti-Wuhan-Hu1 and anti-Delta neutralization capacities resulted in coefficients of 0.91 for the homologous prime-boost AZD1222 group and 0.80 for both the heterologous and the homologous BNT162b2 groups. Moreover, males homologously primed and boosted with AZD1222 seemed to have superior neutralizing capacity compared to females, and there was again a weak negative correlation between neutralization capacity and age for participants immunized with BNTb162b2 only (Appendix A).

### 3.3. All Vaccination Regimen Yielded Significant T Cell Responses

Finally, we compared the various vaccination groups for their T lymphocytes´ capacity to respond to SARS-CoV-2 spike peptide pool stimulation via IFNγ secretion. Figure 3 presents the absolute numbers of spot forming cells, each derived from 500,000 peptide-stimulated PBMCs. One-sample t-tests comparing these absolute numbers to the mean + 2 SEM of spot-forming cells observed in the absence of any stimulation showed a significant increase in all three vaccination groups. These results indicate that both homologous and heterologous prime-boost vaccination regimens induced strong T cell responses. We neither observed an impact of age nor of sex (Appendix A).

## 4. Discussion

Here, we compared humoral and cellular immune responses six months into homologous or heterologous prime boost immunization with AZD1222 and BNT162b2, respectively. We showed that the binding antibody units (BAUs) of anti-Spike IgG were highest in the heterologous prime-boost group, reaching values as high as 24,762 BAU/mL. Thereby, our findings are in line with several previous studies indicating superior immunogenicity following heterologous vaccination [16,17,18]. However, whereas previous studies looked at immune responses as early as two or three weeks after the boost, corresponding to three to four months after primary immunization, here we concentrated on a later time point: six months after priming. Considering that (i) homologous prime-boost vaccination with BNT162b2 have been described to result in peak antibody levels of several thousand BAU/mL between days 14 and 28 after priming and that (ii) the half-life of spike-specific antibodies was estimated to be 55 days [19,20], our median levels of 695 BAU/mL at six months appear to be in line. More impressive are our findings of a median as high as 2177 BAU/mL for the heterologously primed and boosted individuals at six months, who we therefore predict to benefit for another six months. Here, we also confirmed that homologous prime-boost vaccination with AZD1222 yielded the lowest antibody levels with a median around 500 BAU/mL at six months and at least four individuals with BAUs below 100 BAU/mL. This finding is intriguing and addresses the possibility of anti-viral vector antibodies interfering with the second immunization.

Even though our three cohorts were comparable in age, they were significantly different with respect to the percentage of male participants. Sex, however, did not impact antibody levels, whereas there seemed to be weakly negative correlations between age and antibody levels and between age and neutralizing capacity after homologous prime-boost immunization with BNT162b2. These latter findings confirm a recent study from Israel, where BNT162b2 is the predominantly administered vaccine and age also correlated negatively with antibody levels and neutralizing capacity [21]. Thus, a reduced humoral immune response at an advanced age is likely to add to the risk of being admitted to the hospital or experiencing a severe course of disease following infection with SARS-CoV-2 [7]. If confirmed in larger studies, these findings will strongly recommend heterologous prime-boost regimens for older vaccinees.

We were intrigued by our findings of comparable neutralization capacities against the Wuhan-Hu-1 and Delta peptides combined with very strong correlations between both, independent of whether antibodies resulted from a homologous or heterologous prime-boost regimen. As we employed surrogate virus neutralization assays with RBD peptides only, our findings suggest that mutations besides the Delta-specific L452R and T478K are responsible for the loss of neutralization described in conventional virus neutralization tests [22,23]. Moreover, our findings attribute a reduced vaccine effectiveness against Delta not only to the loss of susceptibility against neutralizing antibodies but also to an altered T cell reactivity and/or increased viral transmission [11,24,25].

Finally, neither antibody levels nor neutralization capacities correlated with the peripheral blood T cell efficiency of IFNγ secretion upon stimulation with spike peptides. In fact, all vaccination regimens yielded significant T cell responses, with the variation among groups obscuring any possible superiority. Interestingly, we repeatedly observed that in the absence of stimulating peptides, there is an elevated mean of IFNγ-producing spots in ADZ122 primed PBMCs compared to BNT162b2 primed PBMCs. By now, we would like to speculate that adenoviral priming leads to ongoing low-threshold IFNγ secretion which does not necessarily result in a particularly robust response towards spike peptides [26].

Our study has strengths and limitations. One limitation relates to missing data on the early immune response towards vaccination, obscuring the possibility of carrying out a longitudinal analysis. However, the use of the WHO’s standards allows for reliable comparisons with published data [13]. Another limitation lies in the ethnic homogeneity of our cohort, which may limit the possibility of universalizing our results. Likewise, as we did not specifically record our participants´ health status, we may have looked at a particularly healthy cohort and therefore cannot make inferences based on our findings about populations with different health profiles. In contrast, the strengths of our study lie (i) in the reasonably sized cohorts, (ii) the relatively late time point of six months after priming, and (iii) the direct comparison of two homologous and one heterologous prime-boost vaccination regimens. In summary, our data show that while the heterologous prime-boost vaccination against SARS-CoV-2 prompted a superior humoral response, the T cell responses found for the various vaccination regimens were comparable.

## 5. Conclusions

This study provided evidence that a heterologous prime boost vaccination led to higher anti-SARS-CoV-2 antibody levels than homologous regimen at six months after priming. Neutralization capacities correlated with antibody levels and were comparable against the Wuhan-Hu-1 and Delta strains. In contrast, T cell responses seemed independent of the vaccination regime and were comparable between homologous and heterologous prime boost vaccinations.

## Figures and Tables

**Figure 1 vaccines-10-00322-f001:**
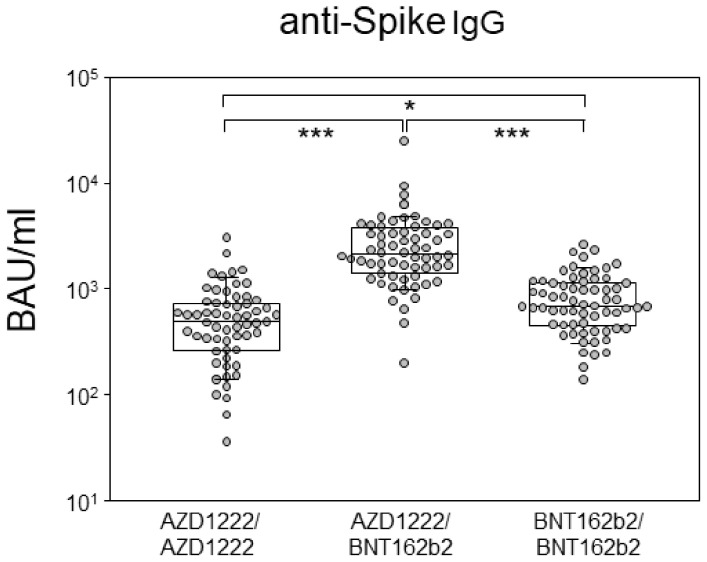
Heterologous prime-boost combining AZD1222 and BNT162b2 resulted in highest antibody levels at six months. Box plots compare anti-spike IgG antibody levels in BAU/mL six months into homologous or heterologous prime-boost vaccinations using AZD1222 and BNT162b2. Each dot represents one individual, while the boxplots show the medians, 25th and 75th percentiles, as well as the 95th percentile via the whiskers. The *p*-value resulting from the Kruskal–Wallis test was < 0.0001, and results from Dunn´s multiple comparisons test are indicated by asterisks and correspond to *p* < 0.05 (*), and *p* < 0.001 (***).

**Figure 2 vaccines-10-00322-f002:**
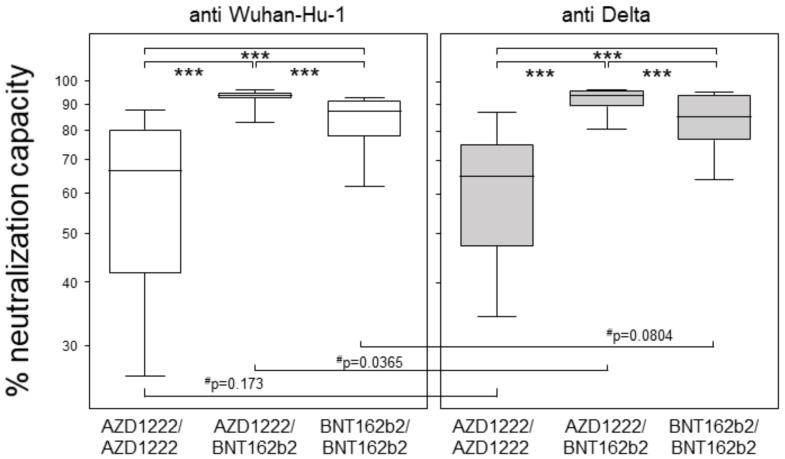
Heterologous prime-boost combining AZD1222 and BNT162b2 yielded highest neutralization capacities. Box plots indicate neutralizing capacities against the Wuhan-Hu-1 strain and the Delta Variant, respectively. Data are shown as percentages of neutralizing antibodies resulting from surrogate virus neutralization tests using RBD peptides. Asterisks indicate significant differences resulting from Kruskal-Wallis (*p* < 0.0001) with Dunn´s multiple comparisons test, with *** representing *p* < 0.001. The pound symbols (^#^) indicate *p*-values resulting from Wilcoxon matched pairs signed rank tests.

**Figure 3 vaccines-10-00322-f003:**
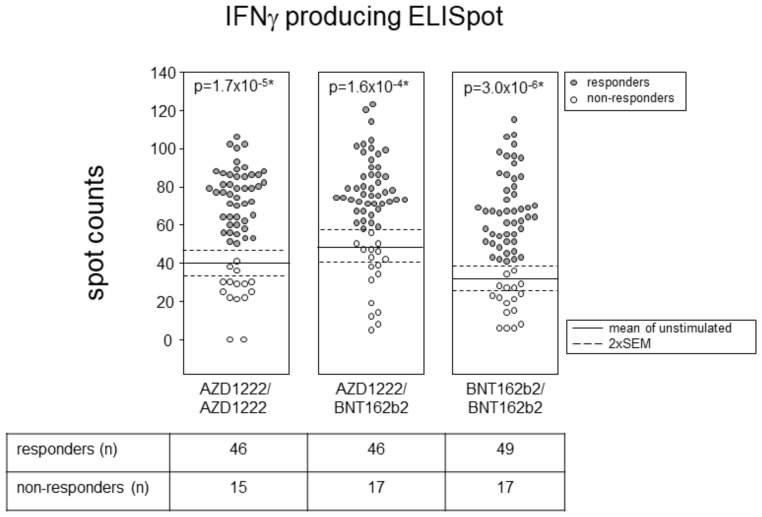
Homologous prime-boost with BNT162b2 yielded the highest T cell responses. Dot plots show absolute numbers of IFNγ-producing ELISpots resulting from 5 × 10^5^ PBMCs stimulated with peptide pools representing the spike protein. Horizontal lines indicate means ± 2 SEM of IFNγ-producing ELISpots present in the absence of peptide stimulation. Each dot represents one individual and, in the case of more than mean + 2 SEM IFNγ-producing ELISpots, individuals were considered responders. *p*-values resulted from one-sample *t*-tests, comparing the numbers of ELISpots after peptide stimulation to the mean + 2 SEM numbers of ELISpots in the absence of stimulation. Pairwise comparisons via Wilcoxon matched pairs of unstimulated and stimulated numbers of ELISpots were 6.07 × 10^−8^ (AZD1222/AZD1222), 8.41 × 10^−7^ (BAZD1222/BNT162b2), and 4.49 × 10^−9^ (BNT162b2/BNT162b2). The table gives numbers of responders and non-responders resulting from each vaccination regimen. A chi-square test for independence confirms comparable proportions with a *p*-value of 0.9546.

**Table 1 vaccines-10-00322-t001:** Patients’ characteristics.

	AZD1222/AZD1222 n = 61	AZD1222/BNT162b2 n = 63	BNT162b2/BNT162b2 n = 66	*p*-Value (Statistical Test)
sex: male/female (n/n)	23/38	11/52	17/49	0.0382 (chi-squared test)
age: median (min-max)	43 (22–65)	42 (20–61)	47 (23–63)	0.1714 (Kruskal-Wallis test)

## Data Availability

Data are contained within the article or Appendix A.

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
