# Peer review of "Higher SARS-CoV-2 Spike Binding Antibody Levels and Neutralization Capacity 6 Months after Heterologous Vaccination with AZD1222 and BNT162b2"

_vaccines, 2022, doi:10.3390/vaccines10020322_

Round 1
Reviewer 1 Report
A well designed, well conducted and well discussed study which small size doesn't affect its strenght and conclusivity. Just a side observation and suggestion: perhaps useful to discuss, in the final chapter, the fact that a presumably healthy sample of health workers was studied, so limiting the possibility to infer the results towards populations with different health profiles.
Author Response
An important point – which is added to our discussion. Please, see page 18
Reviewer 2 Report
This is an important study on heterologous vaccination with AZD1222 and BNT162b2, licensed in European countries. In addition, a comparison of humoral and cellular immune responses at 6 months (a relatively longer-term) after vaccination added an extra value to this study. The manuscript could be further improved if the authors address the following concerns.
- The inclusion of more details on the study population is encouraged. “61 of our participants had received homologous prime-162 boost immunizations with AZD1222 at 10 to 12 weeks intervals (Vaxzevria / AstraZeneca), 66 had received homologous prime-boost immunizations with BNT162b2 at 4 weeks intervals (Comirnaty / Biontech), and 65 participants had been subject to heterologous prime-boost immunization with AZD1222 followed by BNT162b2 at 11 to 12 weeks intervals” is stated. However, the interval between first and second doses is well known to affect the response. It is desired to include the mean and range of the interval for each population as supplementary data.
- In addition, the race and ethnic group of the study population need to be clarified if such data are available.
- In Abstract, in line 25, there is a typo …” respectively. 5Likewise, the neu-“…
- Inclusion of discussion on why homologous prime-boost vaccination with AZD1222 yielded the lowest antibody levels is desired (e.g., a potential role of anti-virus vector antibody).
Author Response
ad 1: We felt that this information is important enough to be added to the results section. So please, see the first paragraph of the results section.
ad 2: As reviewer 1 encouraged us also to elaborate on our study cohort, we added some more information to the materials and methods section (first paragraph) and a couple of sentences to the last paragraph of the discussion. Please see pages 5 and 18.
ad 3: Thanks for pointing that out to us - eliminated
ad 4: A point now added to our discussion. Please see end of first paragraph of the discussion on page 16.
Reviewer 3 Report
The results indicated that heterologous prime-boost vaccination prompted a superior humoral response compared with homologous vaccination while T cell responses as INF gamma production was comparable between the 3 vaccination regimen groups. This is a piece of important information worthwhile for publication
Author Response
thanks!